# Biomarkers representing key aging-related biological pathways are associated with subclinical atherosclerosis and all-cause mortality: The Framingham Study

Cecilia Castro-Diehl[1], Rachel Ehrbar[2], Vanesa Obas[3], Albin Oh[3], Ramachandran S. Vasan[1,4,5,6], Vanessa Xanthakis[1,2,4]*

1 Department of Medicine, Section of Preventive Medicine and Epidemiology, Boston University School of Medicine, Boston, Massachusetts, United States of America, 2 Department of Biostatistics, Boston University School of Public Health, Boston, Massachusetts, United States of America, 3 Department of Medicine, Internal Medicine Residency Program, Boston University School of Medicine, Boston, Massachusetts, United States of America, 4 Lung, and Blood Institute's Framingham Heart Study, Boston University's and National Heart, Framingham, Massachusetts, United States of America, 5 Department of Medicine, Section of Cardiology, Boston University School of Medicine, Boston, Massachusetts, United States of America, 6 Department of Epidemiology, Boston University School of Public Health, Boston, Massachusetts, United States of America

* vanessax@bu.edu

**Data Availability Statement:** All data are available at: https://biolincc.nhlbi.nih.gov/studies/framcohort/ https://biolincc.nhlbi.nih.gov/studies/

## Abstract

### Background

Increased oxidative stress, leukocyte telomere length (LTL) shortening, endothelial dysfunction, and lower insulin-like growth factor (IGF)-1 concentrations reflect key molecular mechanisms of aging. We hypothesized that biomarkers representing these pathways are associated with measures of subclinical atherosclerosis and all-cause mortality.

### Methods and results

We evaluated up to 2,314 Framingham Offspring Study participants (mean age 61 years, 55% women) with available biomarkers of aging: LTL, circulating concentrations of IGF-1, asymmetrical dimethylarginine (ADMA), and urinary F2-Isoprostanes indexed to urinary creatinine. We evaluated the association of each biomarker with coronary artery calcium [ln (CAC+1)] and carotid intima-media thickness (IMT). In multivariable-adjusted linear regression models, higher ADMA levels were associated with higher CAC values ($\beta_{ADMA}$ per 1-SD increase 0.25; 95% confidence interval [CI] [0.11, 0.39]). Additionally, shorter LTL and lower IGF-1 values were associated with higher IMT values ($\beta_{LTL}$ −0.08, 95%CI −0.14, −0.02, and $\beta_{IGF-1}$ −0.04, 95%CI −0.08, −0.01, respectively). During a median follow-up of 15.5 years, 593 subjects died. In multivariable-adjusted Cox regression models, LTL and IGF-1 values were inversely associated with all-cause mortality (hazard ratios [HR] per SD increase in biomarker, 0.85, 95% CI 0.74–0.99, and 0.90, 95% CI 0.82–0.98 for LTL and IGF-1, respectively). F2-Isoprostanes and ADMA values were positively associated with all-cause mortality (HR per SD increase in biomarker, 1.15, 95% CI 1.10–1.22, and 1.10, 95% CI, 1.02–1.20, respectively).

framoffspring/ https://biolincc.nhlbi.nih.gov/studies/gen3/.

**Funding:** Framingham Heart Study (FHS) acknowledges the support of contracts NO1-HC-25195, HHSN268201500001I and 75N92019D00031 from the National Heart, Lung and Blood Institute for this research. This work was also supported by the National Heart, Lung and Blood Institute's 2K24 HL04334 (RSV), 6R01-NS 17950, R01 AG021593, 1RO1-HL64753, R01-HL076784), and RO1HL080124 (RSV), and 1R38HL143584; NIH Boston University Cardiovascular Center, N01-HV- 28178 and NIH grant HL71039 (RSV). This work was also supported by the National Institute on Aging (1R01-AG028321). Dr. Vasan is supported in part by the Evans Medical Foundation and the Jay and Louis Coffman Endowment from the Department of Medicine, Boston University School of Medicine. CCD was supported by the Multidisciplinary Training Program (T32) in Cardiovascular Epidemiology (5T32HL125232). The funders had no role in study design, data collection and analysis, decision to publish, or preparation of the manuscript.

**Competing interests:** The authors have declared that no competing interests exist.

## Conclusion

In our prospective community-based study, aging-related biomarkers were associated with measures of subclinical atherosclerosis cross-sectionally and with all-cause mortality prospectively, supporting the concept that these biomarkers may reflect the aging process in community-dwelling adults.

## Introduction

Advances in the diagnosis and treatment of cardiovascular disease (CVD) have likely contributed to the increased longevity of the U.S. population [1,2]. Yet, although prevalence of CVD has decreased over the course of the past decade, CVD remains the leading cause of death globally [3]. Aging lowers the threshold for susceptibility to CVD by weakening cardioprotective mechanisms [4], increasing stiffness and decreasing distensibility of the vasculature as well as dysregulation of redox balance mechanisms resulting in higher level of oxidants [5] that may lead to adverse cardiovascular remodeling [6]. Several biomarkers representing aging mechanisms have been associated with increased risk of CVD, CVD-related mortality, and all-cause mortality. The major molecular mechanisms of aging-related CVD morbidity include increased oxidative stress [6], telomere attrition [7], endothelial dysfunction [8], mitochondrial autophagy [9], and alterations in concentrations of insulin-like growth factor [10]. Given the predisposition to CVD associated with aging [11], and that the prevalence of subclinical atherosclerosis increases with age, it is conceivable that aging-related biomarkers may be associated with coronary artery calcium (CAC) and carotid intima-media thickness (IMT), which are validated markers of subclinical atherosclerosis [12]. A link between measures of subclinical atherosclerosis and age-related biomarkers has not been previously established and elucidation of such a relation could have important implications in CVD risk stratification. In the present investigation, we hypothesized that biomarkers representing multiple key molecular mechanisms of aging are associated with measures of subclinical atherosclerosis cross-sectionally, and with all-cause mortality prospectively. We tested this hypothesis in a community-based sample comprised of middle-aged adults.

## Methods

### Study sample

In the present investigation, we included participants from the Framingham Heart Study (FHS) Offspring cohort who attended their sixth (1995–1998) and seventh examination cycles (1998–2001). Blood samples for analysis of leukocyte telomere length (LTL) and asymmetrical dimethylarginine (ADMA) were collected during examination cycle 6 and plasma Insulin-like growth factor 1 (IGF-1) and urinary F2-isoprostane 8-iso-prostaglandin (F2-Isoprostanes) were assayed at examination cycle 7. Out of the 3,264 eligible participants who attended both examinations, we excluded participants who did not have available data on blood ADMA (n = 92) and IGF-1 concentrations (n = 421) or urinary F2-Isoprostane (n = 437) levels, resulting in a sample size of 2,314 participants (**Sample 1**). Among 2,314 eligible participants, 1,697 participants had available information on carotid artery intima-media thickness (IMT) (**Sample 2**), and 890 participants had available information on coronary artery calcium (CAC) (**Sample 3**). Finally, among participants from **Sample 1**, 928 participants had available LTL measurements (**Sample 4**). Of note, among the 928 participants with LTL measurements available, 670 had data on IMT, and 253 had data on CAC. IMT and CAC values were measured at

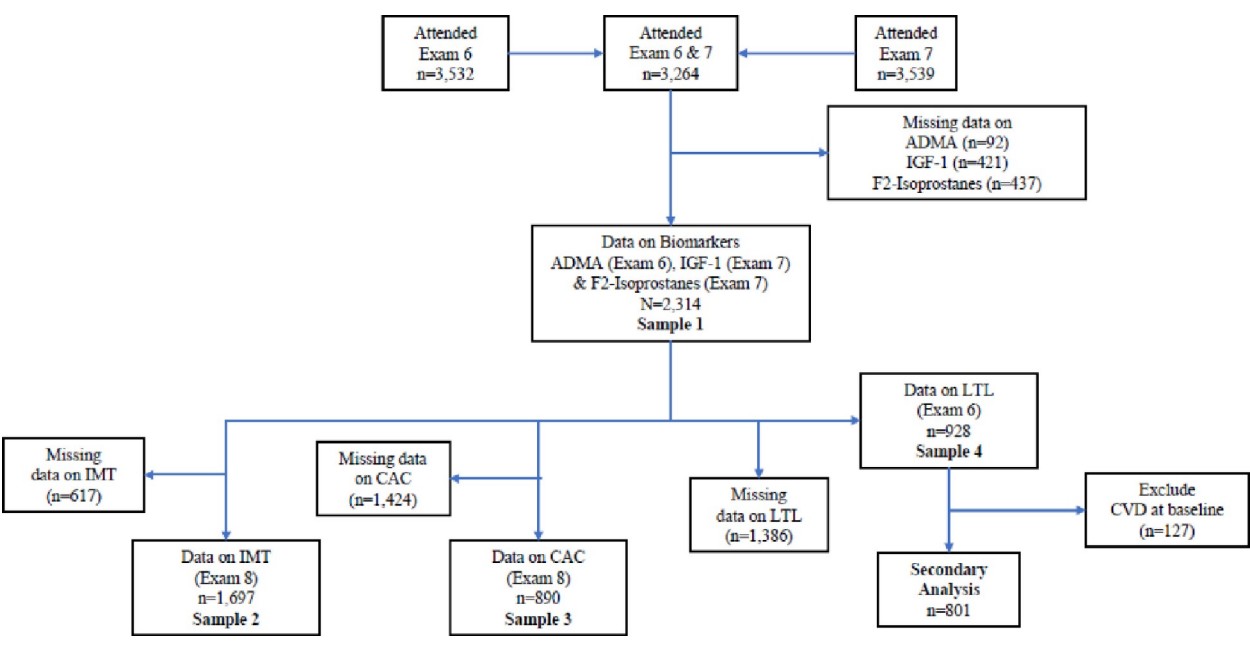

**Fig 1. Derivation of study samples for different analyses.**

examination 8 (2005–2008) and 2002–2005, respectively. In secondary analysis, we removed participants from **Sample 4** with prevalent CVD at baseline (n = 127) and obtained a final sample of 801 participants for this analysis. **Fig 1** depicts the derivation of study samples for different analyses.

The Framingham Heart Study protocol was approved by the Institutional Review Board of Boston University Medical Center, and all participants gave written informed consent at each examination.

## Measurements of aging biomarkers

Participants had blood drawn during their routine examination visit in the morning after a 12-hour overnight fast. After blood was collected, it was immediately centrifuged and then stored at -80˚C until biomarkers were assayed. For this investigation, we used the following biomarkers: LTL, IGF-1, plasma ADMA, and urinary F2-Isoprostanes indexed to urinary creatinine. LTL was assessed by terminal restriction fragment (TRF) length, which was measured by Southern blot analysis. Handling and processing of the samples to analyze TRF length at FHS have been reported elsewhere [13]. The laboratory conducting the measurements was blinded to participants' clinical information. After obtaining the measurements, the laboratory transmitted the data electronically to the FHS data center. Circulating serum concentrations of IGF-1 were analyzed by using commercial assays (R&D Systems, Inc, Minneapolis, MN). Inter-assay coefficient of variation (CV) and intra-assay CV of IGF-1 were 4.5% and 3.4%, respectively. ADMA concentrations were measured from participants' plasma samples stored for 8 years at -80˚C. Assessment of ADMA was performed using a validated high-throughput liquid chromatography–tandem mass spectrometric assay. CV of ADMA was 3.2%. Details of the assays have been described previously [14]. Urinary F2-Isoprostanes is an indicator of systemic oxidative stress. Urine samples were collected during the participant's routine seventh examination cycle and stored at -80˚C until analysis. Assessment of F2-Isoprostanes was performed using a commercially available ELISA (Cayman, Ann Arbor, MI). Samples were

analyzed with an average intra-assay CV of 9.7% and levels were indexed to urinary creatinine and expressed as ng/mmol creatinine. Urinary creatinine had an average intra-assay CV of 2%. Additional details of the processing and analysis of F2-Isoprostanes at FHS are described elsewhere [15].

## Measurements of components of subclinical atherosclerosis

**Coronary artery calcium (CAC) at Offspring eighth examination cycle.** A subsample of FHS participants underwent a chest CT at the 8[th] examination cycle using a multi-slice multi-detector CT scanner (LightSpeed Ultra; General Electric Milwaukee, WI) for the assessment of coronary artery calcium (CAC). Participants were scanned twice by trained technicians following an established protocol, and then examined by experienced readers who identified calcification along the course of the coronary arteries, as reported previously [16]. CAC scores were calculated from the two scans based on a modified Agatston score [17].

**Carotid Intima Media Thickness (IMT) at Offspring eighth examination cycle.** FHS study participants underwent carotid ultrasonography of both right and left carotid arteries at the 8[th] examination cycle using a Toshiba SSH-140A imaging machine as previously described [18]. Briefly, two measurements of carotid arteries were performed using longitudinal views of both the common carotid artery (CCA) and internal carotid artery (ICA) by a trained and certified sonographer, in a standardized protocol, and then reviewed by a radiologist, both of them blinded to participants' clinical information. To assess the intima-media thickness (IMT), the sonographer manually traced intima-media interface lines to quantify the degree of thickening of the carotid artery walls. The mean IMT of the CCA was measured over a segment of 1 cm long, located approximately at 5 mm proximal to the common carotid bulb, so CCA-IMT was defined as the mean of the maximal IMT measurement for the right and left CCA. The maximum IMT of the ICA was defined as the maximum wall thickness in either the right or left ICA extending from the bulb to 10 mm above the carotid sinus, so ICA-IMT was defined as the mean of the maximal IMT measurements for the carotid artery bulb and the ICA on the right and left sides [19]. For this investigation, we defined overall IMT of the CCA and ICA as the mean of the mean IMT values of right and left sides after standardization.

## Outcome of interest

Our primary outcome of interest was all-cause mortality. A group of three investigators examined all hospitalization and physician office records and death certificates to ascertain the cause of death. In secondary analysis, we investigated the association of biomarkers of aging with time to CVD (including coronary heart disease, stroke or transient ischemic attack, heart failure, and intermittent claudication).

## Covariates

During their FHS examination visits, participants underwent a physical examination, responded to a set of questionnaires related to their health status and had laboratory testing for standard CVD risk factors. Covariates included in analyses relating biomarkers of aging to components of subclinical atherosclerosis are as follows: age, sex, body mass index (BMI), current smoking status, systolic blood pressure (SBP), hypertension treatment, diabetes, and total cholesterol/HDL. BMI (kg/m$^2$) was calculated by dividing the weight in kilograms by the square of height in meters. Smoking status (yes/no, in the year preceding the Heart Study examination) was assessed via a self-administered questionnaire. Blood pressure was obtained as the average of two physician-obtained readings taken on seated participants during the FHS examination visit using a standardized protocol. Hypertension medication use was ascertained

from information of medication brought to the examination visit by participants. Diabetes status was defined as fasting glucose $\geq$126 mg/dL or use of treatment (either insulin or a hypoglycemic agent). For the analysis of time to all-cause mortality, in addition to the above covariates, we also adjusted for estimated glomerular filtration rate (eGFR) calculated using the Chronic Kidney Disease (CKD) EPI equation for Glomerular Filtration Rate [20].

## Statistical analysis

**Association of biomarkers with components of subclinical atherosclerosis.** We natural logarithmically-transformed values of CAC to normalize their skewed distributions; ln(CAC +1) was used as the dependent variable in analyses. We standardized the average of the values of the CCA and ICA IMT, and obtained an average IMT for both carotid arteries.

We used multivariable linear regression models to evaluate the association of the biomarkers of aging (independent variables, separate model for each) with CAC and IMT (dependent variables, separate models for each). We initially adjusted all models for age and sex, and then further adjusted for BMI, SBP, hypertension medication use, diabetes, smoking status, and total cholesterol/HDL ratio. We also evaluated potential effect modification of the relation between biomarkers of aging and components of subclinical atherosclerosis by sex by including corresponding interaction terms in the models. A P-value <0.05 for the interaction was considered statistically significant.

**Association of biomarkers with all-cause mortality and CVD.** Follow-up time started after the 7th examination cycle, which served as the baseline for this analysis. After confirming that the assumption of proportionality of hazards was met, we used multivariable Cox regression to relate biomarkers of aging (independent variables, separate model for each) to time to death (dependent variable) adjusting for age, sex, BMI, SBP, hypertension medication use, diabetes, current smoking status, total cholesterol/HDL ratio, and eGFR. To evaluate the conjoint association of the biomarkers of aging with time to death, we included IGF-1, ADMA, and F2-Isoprostanes together in a stepwise regression model using an entry and retention criterion of 0.1 for statistical significance level and forced age, sex, smoking status, BMI, SBP, hypertension medication, diabetes, total cholesterol/HDL ratio, and eGFR into the model. LTL was excluded from the stepwise selection model to retain a larger sample size. We allowed for different baseline hazards by stratifying models according to prevalent CVD status in all Cox regression models.

**Creation of a biomarker score.** We created a biomarker score as follows: first, we used a stepwise selection process in the Cox regression model including three biomarkers (IGF-1, ADMA, and F2-Isoprostanes; LTL was not included to retain a larger sample size). Then, we multiplied the beta estimate for each biomarker obtained by the stepwise model with the individual's biomarker value/concentration (e.g. $\beta_{ADMA}{}^*ADMA$). This created three new variables (products) for each participant which we summed to get the biomarker score, i.e. score = $\beta_{ADMA}{}^*ADMA+\beta_{IGF-1}{}^*IGF1+\beta_{F2}{}^*F2$. The biomarker score was categorized as tertiles in ascending order (tertile 1 with the lowest values vs. tertile 3 with the highest values of the biomarker score). We created a Kaplan-Meier survival curve to graphically present the survival time according to tertiles of the biomarker score. In secondary analysis we created a biomarker score including all four biomarkers (IGF-1, ADMA, F2-Isoprostanes, and LTL) using a subsample including data from participants with available data on all 4 biomarkers.

## Secondary analysis

In secondary analyses, we evaluated the association of aging biomarkers with time to CVD (dependent variable). All biomarkers were included in the same model, also adjusting for the same covariates as in primary analysis for time to death.

# Results

## Baseline characteristics

Baseline characteristics of our study sample are presented in **Table 1**. Our sample had a mean age of 61 years, with an age range of 33 to 88 years, and included 55% women. Among participants with CAC data, 81% of men and 56% of women had CAC score greater than zero.

## Relations of biomarkers of aging with subclinical atherosclerosis

In multivariable-adjusted models, higher ADMA levels were associated with higher CAC scores, while levels of LTL, IGF-1 and F2-Isoprostanes were not significantly associated with CAC (**Table 2**). We observed an inverse association of LTL and IGF-1 concentration with mean IMT values. Levels of ADMA and F2-Isoprostane values were not significantly associated with IMT values (**Table 2**).

**Table 1. Characteristics of study sample.**

| | Men (n = 1038) | Women (n = 1276) |
|---|---|---|
| **Clinical Characteristics** | | |
| Age, y | 61±10 | 61±10 |
| Body mass index, kg/m$^2$ | 28.9±4.6 | 27.6±5.7* |
| Smoking, % | 13 | 12 |
| Diabetes mellitus, % | 14 | 9* |
| Systolic blood pressure, mm Hg | 129±18 | 127±20* |
| Diastolic blood pressure, mm Hg | 76±10 | 73±10* |
| Prevalent hypertension, % | 37 | 31* |
| Hypertension treatment, % | 30 | 25* |
| Serum creatinine, mg/100ml | 1.3±0.2 | 1.1±0.2* |
| Total cholesterol, mg/dL | 192±35 | 207±36* |
| HDL cholesterol, mg/100ml | 46±12 | 61±17* |
| LDL cholesterol, mg/100ml | 119±32 | 120±34 |
| Triglycerides, mg/100ml | 145±106 | 131±76* |
| eGFR, mL/min/1.73m$^2$ | 83±16 | 83±17 |
| **Biomarkers,** median (Q1,Q3) | | |
| Telomere lenght (KB) | 6.9 (6.5, 7.3) | 7.0 (6.6, 7.4)* |
| IGF-1 (ng/ml) | 117 (95, 138) | 100 (81, 124) * |
| ADMA (umol/L) | 0.54 (0.47, 0.62) | 0.53 (0.46, 0.60) * |
| F2-Isocreatinine (ng/mmol/creatinine) | 122.5 (86.6, 177.9) | 146.3 (95.5, 213.2) * |
| **Subclininical Cardiovascular Disease** | | |
| **Coronary artery calcification (CAC)** | | |
| Coronary Artery Calcium Score Median (Q1,Q3) | 130 (5, 513) | 4 (0, 82) * |
| Prevalence of CAC Score[1], % | | * |
| 0 | 19 | 44 |
| 1–100 | 28 | 34 |
| ≥101 | 53 | 22 |
| **Carotid IMT** | | |
| CCA, mm Median (Q1,Q3) | 0.7 (0.6, 0.8) | 0.6 (0.6, 0.7) * |
| ICA, mm Median (Q1,Q3) | 2.4 (1.6, 3.4) | 1.9 (1.3, 2.8) * |

All values shown are mean ± standard deviation or median (Q1, Q3), unless otherwise specified.

[1]Percent is out of those with available CAC score.

*Significant difference (p<0.05) between men and women.

**Table 2. Association of individual biomarkers of aging with components of subclinical atherosclerosis.**

| Biomarker* | Unadjusted model | | Model 1 | | Model 2 | |
|---|---|---|---|---|---|---|
| Association with CAC† | Estimate (95% CI) | p-value | Estimate (95% CI) | p-value | Estimate (95% CI) | p-value |
| LTL (Kb), n = 253 | -0.63 (-0.95, -0.31) | 0.0002 | -0.22 (-0.5, 0.06) | 0.12 | -0.24 (-0.51, 0.04) | 0.09 |
| IGF-1 (ng/ml), n = 890 | -0.07 (-0.25, 0.10) | 0.40 | -0.01 (-0.15, 0.13) | 0.91 | 0.03 (-0.12, 0.17) | 0.71 |
| ADMA (umol/L), n = 890 | 0.48 (0.30, 0.65) | < .0001 | 0.25 (0.11, 0.40) | 0.0006 | **0.25 (0.11, 0.39)** | **0.001** |
| F2-Isoprostane (ng/mmol), n = 890 | -0.10 (-0.29, 0.10) | 0.33 | 0.16 (0.0, 0.32) | 0.05 | 0.06 (-0.10, 0.22) | 0.47 |
| Association with IMT‡ | | | | | | |
| LTL (Kb), n = 670 | -0.22 (-0.29, -0.16) | < .0001 | -0.09 (-0.15,-0.03) | 0.01 | **-0.08 (-0.14, -0.02)** | **0.01** |
| IGF-1 (ng/ml), n = 1697 | -0.08 (-0.12, -0.04) | < .0001 | -0.06 (-0.1,-0.02) | <0.0001 | **-0.04 (-0.08, -0.01)** | **0.02** |
| ADMA (umol/L), n = 1697 | 0.10 (0.06, 0.14) | < .0001 | 0.06 (0.03,0.1) | <0.0001 | 0.02 (-0.02, 0.05) | 0.34 |
| Isoprostane (ng/mmol), n = 1697 | 0.02 (-0.02, 0.07) | 0.27 | 0.03 (-0.01,0.07) | 0.12 | 0.02 (-0.02, 0.06) | 0.24 |

*Biomarkers were analyzed in separate models.

† CAC was modeled as ln(CAC+1).

‡ IMT was modeled as mean of standardized CCA IMT and ICA IMT.

Model 1 is adjusted for age and sex.

Model 2 is adjusted for age, sex, BMI, SBP, hypertension medication, diabetes, current smoking status, and total cholesterol/HDL.

Estimates are per 1 standard deviation increase in the biomarker.

We did not observe significant effect modifications of the relation between biomarkers of aging and subclinical disease by sex (all p values exceeded 0.15).

## Associations of biomarkers of aging with all-cause mortality

During a median follow-up period of 15.5 years, there were 593 deaths (274 in women). In analyses of individual biomarkers, after multivariable adjustment, LTL and IGF-1 values were inversely related whereas ADMA and urinary F2-Isoprostanes concentrations were directly related to risk of death (**Table 3**). When we included all three biomarkers (IGF-1, ADMA and F2-Isoprostanes) in a single model, all biomarkers were significantly associated with all-cause mortality (**Table 4**). The effect size of the jointly modeled three biomarkers and all-cause mortality was similar to those when each biomarker was individually modeled.

**Fig 2A and 2B** depict the comparison of survival time according to tertiles of the biomarker score. The three curves are statistically different (Log-Rank p<0.0001) with tertile 3 having the highest survival compared to tertile 1.

**Table 3. Association of individual biomarkers of aging with all-cause mortality.**

| Biomarker* | HR (95% CI) | p-value |
|---|---|---|
| LTL (Kb) | 0.85 (0.74–0.99) | 0.03 |
| IGF-1 (ng/ml) | 0.90 (0.82–0.98) | 0.02 |
| ADMA (umol/L) | 1.10 (1.02–1.20) | 0.02 |
| F2-Isoprostane (ng/mmol) | 1.15 (1.10–1.22) | <0.0001 |

* Biomarkers were analyzed in separate models.

Models are adjusting for age, sex, BMI, SBP, hypertension medication, diabetes, current smoking status, total cholesterol/HDL, and eGFR.

Hazard ratios are per 1 standard deviation increase in the biomarker.

Sample size: n = 2314 except for LTL (n = 928).

**Table 4. Joint association of biomarkers of aging with all-cause mortality.**

| Biomarker* | HR (95% CI) | p-value |
| --- | --- | --- |
| IGF-1 (ng/ml) | 0.91 (0.83–0.99) | 0.04 |
| ADMA (umol/L) | 1.10 (1.01–1.19) | 0.03 |
| Isoprostane (ng/mmol) | 1.14 (1.08–1.21) | <0.0001 |

*Model forced in covariates and used an entry and stay p-value 0.1.

Note: All three biomarkers were included in one model. Models are adjusted for age, sex, BMI, SBP, hypertension medication, diabetes, current smoking status, total cholesterol/HDL, and eGFR.

Hazard ratios are per 1 standard deviation increase in the biomarker.

Sample size: n = 2314.

## Secondary analysis

During a follow-up period of 15.5 years, there were 160 CVD events. After multivariable adjustment, ADMA and F2-Isoprostanes were positively related to risk of CVD, LTL was inversely associated, whereas IGF-1 was not associated with incident CVD (**S1 Table**).

**Discussion.** *Principal findings.* We observed that higher ADMA levels were associated with higher CAC scores, whereas shorter LTL and lower IGF-1 values were associated with higher IMT values cross-sectionally, adjusting for age, sex, and other cardiovascular risk factors. Shorter LTL and lower IGF-1 concentrations, and higher levels of ADMA and F2-Isoprostanes were associated with higher risk of all-cause mortality prospectively when modeled individually. In addition, IGF-1, ADMA and F2-Isoprostanes levels were jointly associated with all-cause mortality when modeled together. In secondary analyses, ADMA and F2-Isoprostanes (joint model) were positively and LTL was inversely associated with risk of incident CVD.

## Comparison with the literature

**LTL and CVD outcomes.** Epidemiological studies in humans have reported an association between shorter LTL and higher risk of CVD [21] and all-cause mortality [22], but effect sizes have been small and directionality of associations has been inconsistent [23]. It is not

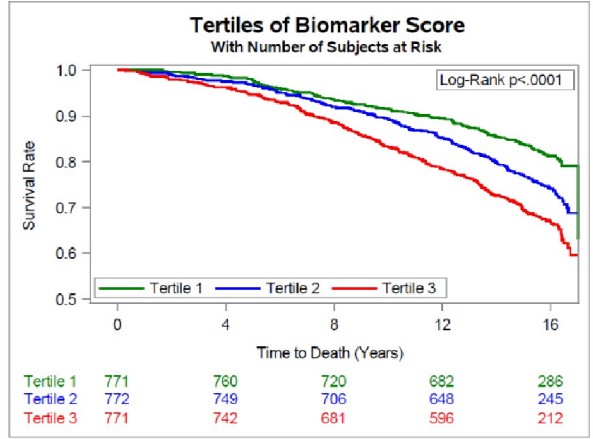
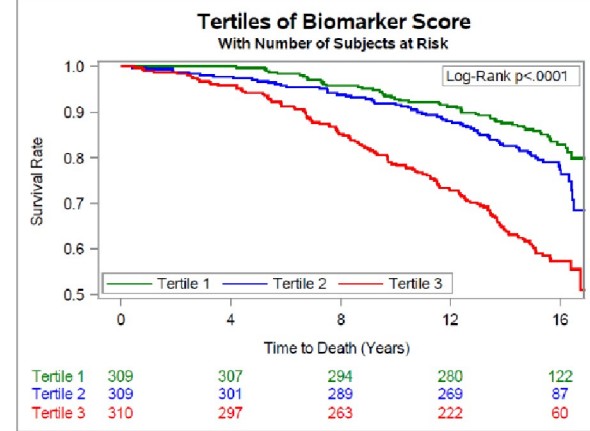

Three biomarkers: Isoprostanes, IGF-1 and ADMA    Four Biomarkers: Isoprostanes, IGF-1, ADMA, and LTL

**Fig 2.** a and b: Kaplan-Meier plots for the relation of the biomarker score including three (panel a) and four (panel b) biomarkers and time to death.

clear whether short LTL is a biomarker, a risk factor, or a consequence of aging, and age-related diseases [24]. It is also possible that LTL shortening is along the causal pathway between oxidative stress and CVD [23].

**LTL and subclinical CVD.** Our findings regarding the association of LTL with IMT are in accordance with other studies. Shorter LTL has been associated with presence of carotid plaque [25], incident CCA-IMT [26], and presence of higher CCA-IMT and ICA-IMT [27]. In the latter study, using data from FHS Offspring participants, the authors observed an inverse association between LTL and ICA-IMT in the overall sample, whereas an association of LTL and CCA-IMT was statistically significant only among obese men [27]. Our investigation differed from this prior study in two respects. One, our participants were at least a decade older than participants from the other FHS investigation. Second, we defined our outcome, carotid IMT, as the average of the means of CCA-IMT and ICA-IMT after standardization. The inverse association of LTL and mean carotid -IMT in our sample remained statistically significant after adjustment for age, sex, BMI and other CVD risk factors. Our finding that LTL is associated with IMT is consistent with most published reports [26,28]. However, consistent with a report by Fernandez-Alvira et al. from the Progression of Early Subclinical Atherosclerosis (PESA) study [29], we did not observe an association between LTL and CAC score. However, in a cross-sectional study of asymptomatic middle-aged adults, shorter LTL was significantly associated with higher CAC score [30]. The latter study differs from ours in that the sample was smaller in size and participants were younger and healthier, but the effect size of the association between LTL and CAC was small [30]. We included a much larger sample of participants, but we did not observe an association between LTL and CAC after adjustment for other variables such as age, smoking, and obesity. Longitudinal studies are warranted to define better whether there is a significant association between shortening LTL and progression of CAC.

**LTL and all-cause mortality.** Previous studies have reported an inverse association between LTL and all-cause mortality consistent with our findings [31–35]. A recent meta-analysis [22] of 25 investigations based on prospective data collection reported an association between shorter LTL and higher risk of all-cause mortality, with a moderate effect size of the association that was similar in the two youngest groups (<75 years and 75–80 years) but weaker in the oldest group (over 80 years). Further subgroup analyses demonstrated that LTL measurement technique, sex, age, ethnicity, and the number of covariates included contributed to the between-study heterogeneity [22]. Similar to previous studies, in our investigation the effect size for the association of LTL with risk of all-cause mortality was modest. It is not yet clear whether shortening LTL triggers a process that will lead to earlier mortality or whether there are other biological processes that cause shortening of LTL [34]. To add to this complex scenario, other factors could interfere in this association like telomerase activity and amount of oxidative stress, which we did not account for in our statistical models.

## IGF-1 and CVD outcomes

In experimental studies with mice, lower IGF-1 concentrations have been associated with extended longevity probably due to lower oxidative stress and ROS production [36]. However, in epidemiological studies in humans the findings have been inconsistent. Lower concentrations of IGF-1 were associated with the development of CHF [37], and higher risks for incident CHD in some reports [38–40] whereas others have reported no association between levels of IGF-1 and incident CVD [41] or CVD mortality [39,42,43]. More recently, few studies have reported a U-shaped association between IGF-1 and CVD mortality [44]. These results may indicate the need for additional studies focusing on earlier stages of atherosclerosis.

**IGF-1 and subclinical CVD.** In concordance with our findings, two other studies [45,46] reported an inverse association of IGF-1 concentrations and carotid IMT. Other epidemiological studies with small-to-moderate sample sizes did not find association between lower IGF-1 concentrations and IMT [47], or observed higher levels of IGF-1 related to higher CCA IMT [48]. However, a study investigating this association, reported a U-shaped association in a small sample of older people without known CVD [49]. In the present investigation, we did not observe any association between IGF-1 and CAC. Studies on the association of low concentrations of IGF-1 with subclinical cardiovascular disease have yielded contradictory results. It is unclear why lower IGF-1 is associated with higher carotid IMT but no with higher coronary calcium in the present investigation. It has been proposed that subclinical disease in the coronary and carotid arteries may reflect different atherosclerotic processes/measures [49].

**IGF-1 and all-cause mortality.** The mechanisms by which a low IGF-1 serum level may be associated with reduced mortality are not clear. It has been proposed that IGF-1 may protect against atherosclerosis (as opposed to the opposite pro-atherosclerotic effects of oxidative stress and inflammation) and increase the production of NO and NOS in endothelial cells. IGF-1 also is involved in the vascular aging process by preventing oxidative smooth muscle cell apoptosis and reducing proinflammatory cytokine production in atherosclerotic plaques [50]. The relation between circulating IGF-1 concentrations and mortality is complex even in epidemiological studies. Low circulating IGF-1 concentrations have been associated with increased risk of cardiovascular mortality in some studies [51,52] but not in others [43]. Circulating concentrations of IGF-1 have also been associated with all-cause mortality, in a U shape manner [41]. In a national sample of more than 20,000 participants that was followed between 4 and 12 years, IGF binding protein (BP)-3 concentrations but not IGF-1 levels was associated with cardiovascular or all-cause mortality [43]. This sample differed from ours in that it is a nationally representative sample and included people 20 years or older. The effect of IGF-1 on aging (and related outcomes) may have been attenuated by including younger people in the latter study. Contrary to the aforementioned national sample, we observed an inverse association between IGF-1 and all-cause mortality in our sample. Our sample was followed for a longer period than the national sample, resulting in a larger number of events and greater statistical power to elucidate associations.

## ADMA and CVD outcomes

ADMA is an indicator of endothelial dysfunction and has been observed to be elevated in people with CVD [53]. ADMA competes with nitric oxide (NO) synthase (NOS), an enzyme that synthesizes endothelial-derived NO, a potent vasodilator [54]. NO deficit may lead to endothelial dysfunction [55]. ADMA also may increase reactive oxygen species (ROS), which cause cell oxidative damage and thereby may play a key role in the aging process [56].

**ADMA and subclinical CVD.** In the current investigation, participants with higher levels of ADMA had higher CAC scores than those with lower levels in accordance with at least two previous studies. [57,58] although these latter studies differed from our investigation in that their sample size was smaller, and included mostly black participants [57] or Japanese patients with CKD [58]. Higher levels of ADMA have been associated with the presence of subclinical atherosclerosis as defined by CAC and carotid IMT; however, we did not observe a significant association between blood ADMA levels and IMT. Contrary to our investigation, two other community-based studies with moderate sample sizes (575 and 922 participants, respectively) observed associations between circulating ADMA concentrations and carotid IMT in a non-white sample [59,60]. In a recent publication using data from FHS Offspring participants, higher levels of blood ADMA were associated with presence of greater ICA/bulb-IMT but not

with CCA-IMT [61]. We assessed carotid-IMT as an average of both ICA-IMT and CCA-IMT, so it is possible that the effect of ICA-IMT may have been attenuated by averaging ICA and CCA.

**ADMA and all-cause mortality.** The association between ADMA and all-cause mortality has been investigated in population-based studies [62] and in clinical studies with CAD [63] or terminal renal disease patients [64], but findings are still inconsistent. In a previous analysis using data from the Framingham Heart Study [62], we observed a significant association of ADMA with all-cause mortality only among non-diabetic participants. In the report by Leong et al., a community-based study of an all-women cohort [64], the risk for all-cause mortality was slightly higher with higher ADMA levels, but did not reach statistical significance after 24 years of follow-up. In the current Framingham sample that includes middle-aged to older men and women with a large follow-up period and more incident fatal events than previous studies, we observed a positive association of ADMA and all-cause mortality.

## F2-Isoprostanes and CVD outcomes

F2-Isoprostanes, a measure of oxidative stress, is another proposed biomarker for aging [65]. Many studies have dealt with the association of F2-Isoprostanes and several medical conditions. In a recent meta-analysis that compares F2-Isoprostanes levels between cases and controls across 50 different health outcomes, F2-Isoprostanes levels were only moderately associated with CHF and ischemic stroke, and weakly associated with CAD, and cancer [66]. However, in larger community-based studies (n = 227 to 8354) F2-Isoprostanes were associated with fatal CHD, but not with nonfatal CHD; we speculate that this finding may suggest that F2-Isoprostanes could be more causally linked with enhanced pathological remodeling, but it warrants further investigation [67,68].

**F2-Isoprostanes and subclinical CVD.** Unlike our study, at least two studies [69,70] observed positive associations of F2-Isoprostane levels with coronary calcification. In one, a community-based study of young black and white subjects, F2-Isoprostanes was assessed in blood, rather than urine [69]. In the other study [70], the sample consisted of a small group of Japanese patients with type 2 diabetes. In a case-control study with a small sample size (n = 30 patients), those with greater carotid or iliofemoral IMT (>0.5mm) had increased urinary levels of F2-Isoprostanes compared to those with lower IMT [71].

Previous evidence has linked F2-Isoprostanes, a marker of oxidative stress, to CVD especially in individuals with clinical CVD. Ours sample was comprised of individuals without overt clinical CVD and with a low prevalence of subclinical CVD; this may explain the lack of statistically significant associations between F2-isoprostane and IMT or CAC in the present investigation.

**F2-Isoprostanes and all-cause mortality.** In the present study, we observed a significant association between urinary F2-Isoprostanes and all-cause mortality. Our finding is in accordance with a large German cohort study of adults (n~8,000) with a follow up of 14 years, in which F2-Isoprostane urinary levels were associated with CVD mortality [68].

## Strength and limitations

The strengths of our investigation are the use of a community-based sample with a wide age range and long-term follow-up. We evaluated a panel of four different biomarkers that represent distinctive biological mechanisms of aging. Participants in our study were very well characterized with measurements of multiple covariates that may be confounders of these associations and we adjusted for these factors in multivariable analyses. Some limitations of our investigation merit consideration. We measured aging biomarkers at only a single time

point, so we could not evaluate the impact of the changes of these biomarkers over time on subclinical CVD or mortality risk. The number of participants with values of telomere length was smaller than the number of participants with available data on the other biomarkers. The assays of all four biomarkers were not concurrent as they were collected over two different sets of FHS examinations. Another limitation of our study is the predominantly white FHS sample, which could limit the generalizability of our findings to other non-white races/ethnicities.

## Conclusion

Our results support the concept that key molecular aging pathways represented by select biomarkers investigated in our study may be markers of mortality risk. The study of the aging process may help reduce age-related disease prevalence and premature mortality. Additional studies of larger multiethnic samples are warranted to confirm our findings.

## Supporting information

**S1 Table. Joint association of biomarkers of aging with incident CVD (including all 4 biomarkers).**
(DOCX)

## Acknowledgments

We acknowledge the dedication of the FHS study participants without whom this research would not be possible.

## Author Contributions

**Conceptualization:** Ramachandran S. Vasan, Vanessa Xanthakis.

**Formal analysis:** Rachel Ehrbar.

**Investigation:** Cecilia Castro-Diehl, Ramachandran S. Vasan, Vanessa Xanthakis.

**Methodology:** Ramachandran S. Vasan, Vanessa Xanthakis.

**Writing – original draft:** Cecilia Castro-Diehl, Vanesa Obas.

**Writing – review & editing:** Albin Oh, Ramachandran S. Vasan, Vanessa Xanthakis.

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
