## [Decision Letter · Decision Letter 0]

16 Nov 2020

PONE-D-20-32445

Biomarkers representing key aging-related biological pathways are associated with subclinical atherosclerosis and all-cause mortality: The Framingham Study

PLOS ONE

Dear Dr. Xanthakis,

Thank you for submitting your manuscript to PLOS ONE. After careful consideration, we feel that it has merit but does not fully meet PLOS ONE’s publication criteria as it currently stands. Therefore, we invite you to submit a revised version of the manuscript that addresses the points raised during the review process.

Castro-Diehl et al in their manuscript entitled "Biomarkers representing key aging-related biological pathways are associated with subclinical atherosclerosis and all-cause mortality: The Framingham Study" tried to investigate the association of leukocyte telomere length (LTL), plasma insulin-like growth factor (IGF-1), asymmetrical dimethylarginine (ADMA) and urine F2-isoprostanes with coronary  artery calcium (CAC), carotid intima-media thickness (IMT) values and with all-cause mortality. It is an interesting study. However, there are some major concerns.

1.     The authors mentioned (line 38) 1244 participants had available LTL measurements, but in figure 1, it showed 1386 LTL measurements were missing. It is also not clear on figure 1 if the 928 participants had the information of CAC and IMT.

2.     Please make statistical comparisons of biomarkers, CAC and IMT between men and women in table 1.

3.     On line 249, Please add references for the sentence “our finding that LTL associated with IMT is consistent with most published report.”

We look forward to receiving your revised manuscript.

Kind regards,

Ying-Mei Feng

Academic Editor

PLOS ONE

Journal Requirements:

3.Thank you for stating the following in the Acknowledgments Section of your manuscript:

[The Framingham Heart Study (FHS) acknowledges the support of contracts NO1-HC-25195,

HHSN268201500001I and 75N92019D00031 from the National Heart, Lung and Blood Institute for this

research. This work was also supported by the National Heart, Lung and Blood Institute's 2K24 HL04334

(RSV), 6R01-NS 17950, R01 AG021593, 1RO1-HL64753, R01-HL076784), and RO1HL080124 (RSV),

and 1R38HL143584; NIH Boston University Cardiovascular Center, N01-HV- 28178 and NIH grants

HL64753 and HL71039 (RSV). This work was also supported by the National Institute on Aging (1R01-

AG028321).Dr. Vasan is supported in part by the Evans Medical Foundation and the Jay and Louis

Coffman Endowment from the Department of Medicine, Boston University School of Medicine. CCD

was supported by the Multidisciplinary Training Program (T32) in Cardiovascular Epidemiology

(5T32HL125232).]

 [NO - The funders had no role in study design, data collection and analysis, decision to publish, or preparation of the manuscript.]

Reviewers' comments:

Reviewer's Responses to Questions

**Comments to the Author**

1. Is the manuscript technically sound, and do the data support the conclusions?

Reviewer #1: Partly

2. Has the statistical analysis been performed appropriately and rigorously? 

Reviewer #1: Yes

3. Have the authors made all data underlying the findings in their manuscript fully available?

Reviewer #1: Yes

4. Is the manuscript presented in an intelligible fashion and written in standard English?

Reviewer #1: Yes

5. Review Comments to the Author

Reviewer #1: Castro-Diehl et al in their manuscript entitled "Biomarkers representing key aging-related biological pathways are associated with subclinical atherosclerosis and all-cause mortality: The Framingham Study" tried to investigate the association of leukocyte telomere length (LTL), plasma insulin-like growth factor (IGF-1), asymmetrical dimethylarginine (ADMA) and urine F2-isoprostanes with coronary artery calcium (CAC), carotid intima-media thickness (IMT) values and with all-cause mortality. It is an interesting study. However, there are some major concerns.

1. The authors mentioned (line 38) 1244 participants had available LTL measurements, but in figure 1, it showed 1386 LTL measurements were missing. It is also not clear on figure 1 if the 928 participants had the information of CAC and IMT.

2. Please make statistical comparisons of biomarkers, CAC and IMT between men and women in table 1.

3. On line 249, Please add references for the sentence “our finding that LTL associated with IMT is consistent with most published report.”

6. PLOS authors have the option to publish the peer review history of their article (what does this mean?). If published, this will include your full peer review and any attached files.

Reviewer #1: No

---

## [Author Response · Author response to Decision Letter 0]

18 Nov 2020

Response to Reviewers

Reviewer #1: 

1. The authors mentioned (line 38) 1244 participants had available LTL measurements, but in figure 1, it showed 1386 LTL measurements were missing. It is also not clear on figure 1 if the 928 participants had the information of CAC and IMT.

Response: We thank the Reviewer for this comment. We now clarify in the revised version of the manuscript that among the participants from sample 1, 928 participants had available LTL measurement (Sample 4).”, as follows:

Page 5, lines 38-39: “Finally, among the participants from Sample 1, 928 participants had available LTL measurements (Sample 4)” 

2. Please make statistical comparisons of biomarkers, CAC and IMT between men and women in table 1.

Response: We thank the Reviewer for this suggestion. We have now performed statistical comparisons between men and women for all variable in Table 1 and included a footnote denoting significant results in the revised Table 1.

3. On line 249, Please add references for the sentence “our finding that LTL associated with IMT is consistent with most published report.”

Response: We regret this oversight. We have now cited two references, as follows:

Page 17, line 251: “Our finding that LTL is associated with IMT is consistent with most published reports. (26, 28)”

---

## [Decision Letter · Decision Letter 1]

19 Jan 2021

PONE-D-20-32445R1

Biomarkers representing key aging-related biological pathways are associated with subclinical atherosclerosis and all-cause mortality: The Framingham Study

PLOS ONE

Dear Dr. Xanthakis,

Thank you for submitting your manuscript to PLOS ONE. After careful consideration, we feel that it has merit but does not fully meet PLOS ONE’s publication criteria as it currently stands. Therefore, we invite you to submit a revised version of the manuscript that addresses the points raised during the review process.

Specifically, Reviewer 2 still has some concerns about the study design and data analysis.  Please submit your revised manuscript by Mar 05 2021 11:59PM. If you will need more time than this to complete your revisions, please reply to this message or contact the journal office at plosone@plos.org. Please include the following items when submitting your revised manuscript:

We look forward to receiving your revised manuscript.

Kind regards,

Yan Li, MD, PhD

Academic Editor

PLOS ONE

Reviewers' comments:

Reviewer's Responses to Questions

**Comments to the Author**

1. If the authors have adequately addressed your comments raised in a previous round of review and you feel that this manuscript is now acceptable for publication, you may indicate that here to bypass the “Comments to the Author” section, enter your conflict of interest statement in the “Confidential to Editor” section, and submit your "Accept" recommendation.

Reviewer #1: All comments have been addressed

Reviewer #2: (No Response)

2. Is the manuscript technically sound, and do the data support the conclusions?

Reviewer #1: Yes

Reviewer #2: Partly

3. Has the statistical analysis been performed appropriately and rigorously? 

Reviewer #1: Yes

Reviewer #2: No

4. Have the authors made all data underlying the findings in their manuscript fully available?

Reviewer #1: Yes

Reviewer #2: Yes

5. Is the manuscript presented in an intelligible fashion and written in standard English?

Reviewer #1: Yes

Reviewer #2: Yes

6. Review Comments to the Author

Reviewer #1: (No Response)

Reviewer #2: In the current article, authors prospectively investigated the associations between aging related biomarkers with subclinical atherosclerosis and all-cause mortality. However, there are some issues need to be noted:

1. Do the 928 patients used for LTL analyses have both IMT and CAC data? Please report the number of deaths in LTL group.

2. IMT and CAC were measured at the eighth follow-up visit, and were not performed simultaneously with the sixth and seventh blood draws and urine retention. The time interval may be more than eight years apart, so they are not cross-sectional analyses.

3. The results of univariate analyses without adjusting confounding factors need to be reported.

4. Table 1 compares the characteristics of the population between sex, so whether the relationships between biomarkers and outcomes are different in male and female?

5. In the analysis of individual biomarkers and mortality risk, each marker has predictive value. So, just because the number of patients with LTL data is small, then taken this parameter out of the subsequent analysis model, and calculated the biomarker score using the other three parameters, is not rigorous.

6. Please clarify the age range of the study population.

7. In the discussion section, the authors repeatedly mentioned the influence of the age of the study population on the research results. So according to the author's understanding, which age group has the most predictive value of the relationship between these biomarkers and outcome events?

7. PLOS authors have the option to publish the peer review history of their article (what does this mean?). If published, this will include your full peer review and any attached files.

Reviewer #1: No

Reviewer #2: No

---

## [Author Response · Author response to Decision Letter 1]

5 Mar 2021

Response to Reviewers

PONE-D-20-32445R1

Biomarkers representing key aging-related biological pathways are associated with subclinical atherosclerosis and all-cause mortality: The Framingham Study

Reviewer #2: In the current article, authors prospectively investigated the associations between aging related biomarkers with subclinical atherosclerosis and all-cause mortality. However, there are some issues need to be noted:

1. Do the 928 patients used for LTL analyses have both IMT and CAC data? Please report the number of deaths in LTL group. 

Response: We thank the Reviewer for this question. We would like to clarify that not all 928 participants included in analyses had available data on both IMT and CAC. Among the 928 participants, 670 had data on IMT, 253 had data on CAC, and 223 participants had data on both IMT and CAC. In addition, there were 250 deaths among the 928 participants used for LTL analyses. We have now added this information in the revised version of the manuscript, as follows:

Page 5, lines 63-65: “Of note, among the 928 participants with LTL measurements available, 670 had data on IMT, 253 had data on CAC, and 223 participants had data on both IMT and CAC. IMT and CAC values were measured at examination 8 (2005-2008) and 2002-2005, respectively.” 

2. IMT and CAC were measured at the eighth follow-up visit, and were not performed simultaneously with the sixth and seventh blood draws and urine retention. The time interval may be more than eight years apart, so they are not cross-sectional analyses. 

Response: We thank the Reviewer for this question. It is correct that not all measures were performed simultaneously and that the time interval between collection of some of the biomarkers and IMT and CAC is more than eight years. ADMA and LTL were assessed at examination cycle 6 (1995-1998), IMT at examination cycle 8 (2005-2008), and CAC during 2002-2005. Since we are not following participants for incidence of events, we have defined this analysis as a cross-sectional investigation. However, if the Reviewer feels strongly, we are happy to remove all mention of “cross-sectional” from the text. We have also added the following in the revised version of the manuscript:

Page 5, lines 64-65: “IMT and CAC values were measured at examination 8 (2005-2008) and 2002-2005 for CAC, respectively.” 

3. The results of univariate analyses without adjusting confounding factors need to be reported. 

Response: To address the Reviewer’s concern, we have now updated Table 2, in which we have included the estimates (and 95% CI) for unadjusted models relating aging biomarkers with IMT and CAC. 

Revised Table 2. Association of individual biomarkers of aging with components of subclinical atherosclerosis.

Biomarker* Unadjusted model Model 1 Model 2 

Association with CAC† Estimate

 (95% CI) p-value Estimate

 (95% CI) p-value Estimate

 (95% CI) p-value

LTL (Kb) -0.63 (-0.95, -0.31) 0.0002 -0.22 (-0.5, 0.06) 0.12 -0.24 (-0.51, 0.04) 0.09

IGF-1 (ng/ml) -0.07 (-0.25, 0.10) 0.40 0.16 (0.0, 0.32) 0.05 0.03 (-0.12, 0.17) 0.71

ADMA (umol/L) 0.48 (0.30, 0.65) <.0001 -0.01 (-0.15, 0.13) 0.91 0.25 (0.11, 0.39) 0.001

F2-Isoprostane (ng/mmol) -0.10 (-0.29, 0.10) 0.33 0.25 (0.11, 0.40) 0.0006 0.06 (-0.10, 0.22) 0.47

Association with IMT‡ 

LTL (Kb) -0.22 (-0.29, -0.16) <.0001 -0.09 (-0.15,-0.03) 0.01 -0.08 (-0.14, -0.02) 0.01

IGF-1 (ng/ml) -0.08 (-0.12, -0.04) <.0001 0.06 (0.03,0.1) <0.0001 -0.04 (-0.08, -0.01) 0.02

ADMA (umol/L) 0.10 (0.06, 0.14) <.0001 -0.06 (-0.1,-0.02) <0.0001 0.02 (-0.02, 0.05) 0.34

Isoprostane (ng/mmol) 0.02 (-0.02, 0.07) 0.27 0.03 (-0.01,0.07) 0.12 0.02 (-0.02, 0.06) 0.24

*Biomarkers were analyzed in separate models

† CAC was modeled as ln(CAC+1)

‡ IMT was modeled as mean of standardized CCA IMT and ICA IMT

Model 1 is adjusted for age and sex.

Model 2 is adjusted for age, sex, BMI, SBP, hypertension medication, diabetes, current smoking status, and total cholesterol/HDL 

Estimates are per 1 standard deviation increase in the biomarker

Sample sizes: nCAC = 890 nIMT = 1697 nLTL = 928

4. Table 1 compares the characteristics of the population between sex, so whether the relationships between biomarkers and outcomes are different in male and female? 

Response: We thank the Reviewer for raising this important question. To address the Reviewer’s comment, we evaluated potential effect modifications of the relation between aging biomarkers and death by sex and display results in Reviewer Table 1 below. We did not observe any significant effect modifications by sex. We also added the following in the revised version of the manuscript:

Page 10, Line 156-159: “We also evaluated potential effect modification of the relation between biomarkers of aging and components of subclinical atherosclerosis by sex by including corresponding interaction terms in the models. A P-value <0.05 for the interaction was considered statistically significant.”

Page 14, Line 220-221: “We did not observe significant effect modifications of the relation between biomarkers of aging and subclinical disease indices by sex (all p values exceeded 0.15)” 

Reviewer Table 1. Evaluation of effect modification of the relation between biomarkers of aging and subclinical disease indices by sex.

Biomarker* Estimate

 (95% CI) p-value

Association with CAC† 

LTL*Male -0.38(-0.93,0.17) 0.17

IGF-1*male 0.05(-0.24,0.34) 0.73

ADMA*male 0.09(-0.19,0.37) 0.52

F2-Isoprostane*male 0.04(-0.32,0.39) 0.84

Association with IMT‡ 

LTL*male 0.01(-0.11,0.13) 0.82

IGF-1*male -0.04(-0.11,0.03) 0.29

ADMA*male -0.01(-0.08,0.06) 0.72

F2-Isoprostane*male -0.05(-0.12,0.02) 0.19

5. In the analysis of individual biomarkers and mortality risk, each marker has predictive value. So, just because the number of patients with LTL data is small, then taken this parameter out of the subsequent analysis model, and calculated the biomarker score using the other three parameters, is not rigorous.

Response: We thank the Reviewer for raising this important point. To address the Reviewer’s comment, we have now created a score that includes all four biomarkers and have added results in Figure 1b and in the text, as follows:

Page 11, lines 184-186: “In secondary analysis we created a biomarker score including all four biomarkers (IGF-1, ADMA, F2-Isoprostanes, and LTL).”

6. Please clarify the age range of the study population.

Response: We thank the Reviewer for this comment. We have now added the age range of the study sample as follows:

Page 12, line 198: “…with an age range of 33 to 88 years,…”

7. In the discussion section, the authors repeatedly mentioned the influence of the age of the study population on the research results. So according to the author's understanding, which age group has the most predictive value of the relationship between these biomarkers and outcome events? 

Response: We thank the Reviewer for raising this point. The focus of our investigation was to evaluate the relation of aging biomarkers with indices of subclinical disease and all-cause mortality. We did not focus on performing prediction models in this investigation. We hope the Reviewer agrees with our approach.

---

## [Decision Letter · Decision Letter 2]

6 Apr 2021

PONE-D-20-32445R2

Biomarkers representing key aging-related biological pathways are associated with subclinical atherosclerosis and all-cause mortality: The Framingham Study

PLOS ONE

Dear Dr. Xanthakis,

Thank you for submitting your manuscript to PLOS ONE. After careful consideration, we feel that it has merit but does not fully meet PLOS ONE’s publication criteria as it currently stands. Therefore, we invite you to submit a revised version of the manuscript that addresses the points raised during the review process.

Please revise the manuscript according to the Reviewer's comments. Especially, please check the accuracy of data shown in tables and figures.

We look forward to receiving your revised manuscript.

Kind regards,

Yan Li, MD, PhD

Academic Editor

PLOS ONE

Journal Requirements:

Reviewers' comments:

Reviewer's Responses to Questions

**Comments to the Author**

1. If the authors have adequately addressed your comments raised in a previous round of review and you feel that this manuscript is now acceptable for publication, you may indicate that here to bypass the “Comments to the Author” section, enter your conflict of interest statement in the “Confidential to Editor” section, and submit your "Accept" recommendation.

Reviewer #2: (No Response)

2. Is the manuscript technically sound, and do the data support the conclusions?

Reviewer #2: (No Response)

3. Has the statistical analysis been performed appropriately and rigorously? 

Reviewer #2: (No Response)

4. Have the authors made all data underlying the findings in their manuscript fully available?

Reviewer #2: (No Response)

5. Is the manuscript presented in an intelligible fashion and written in standard English?

Reviewer #2: (No Response)

6. Review Comments to the Author

Reviewer #2: In this revised manuscript, authors addressed most of the concerns, only some small details need to be improved.

1. In table2, since the sample size differed from each biomarker, especially for LTL, the study sample size should be labeled after the biomarkers.

2. When considered LTL in the multivariate model, the sample size of Table S1 should not be 2010.

7. PLOS authors have the option to publish the peer review history of their article (what does this mean?). If published, this will include your full peer review and any attached files.

Reviewer #2: No

---

## [Author Response · Author response to Decision Letter 2]

8 Apr 2021

Response to Reviewers

PONE-D-20-32445R2

Biomarkers representing key aging-related biological pathways are associated with subclinical atherosclerosis and all-cause mortality: The Framingham Study

Reviewer #2: 

1. In Table 2, since the sample size differed from each biomarker, especially for LTL, the study sample size should be labeled after the biomarkers.

Response: We thank the Reviewer for this point. We have now added the sample size after each biomarker in Table 2.

2. When considered LTL in the multivariate model, the sample size of Table S1 should be 2010.

Response: We thank the Reviewer for this comment, we have now revised Table S1 to address the comment.

---

## [Editor Report · Decision Letter 3]

26 Apr 2021

Biomarkers representing key aging-related biological pathways are associated with subclinical atherosclerosis and all-cause mortality: The Framingham Study

PONE-D-20-32445R3

Dear Dr. Xanthakis,

We’re pleased to inform you that your manuscript has been judged scientifically suitable for publication and will be formally accepted for publication once it meets all outstanding technical requirements.

Kind regards,

Yan Li, MD, PhD

Academic Editor

PLOS ONE